# Application of The FODMAP Diet in a Paediatric Setting

**DOI:** 10.3390/nu14204369

**Published:** 2022-10-18

**Authors:** Dakota Rhys-Jones, Jane E. Varney, Jane G. Muir, Peter R. Gibson, Emma P. Halmos

**Affiliations:** Department of Gastroenterology, Central Clinical School, Monash University and Alfred Health, Melbourne, VIC 3004, Australia

**Keywords:** disorders of gut–brain interaction, irritable bowel syndrome, diet therapy, disordered eating, nutrition

## Abstract

In adults, dietary management, particularly with the FODMAP diet, is a key evidence-based part of multimodal therapy for patients with disorders of the gut–brain interaction, particularly irritable bowel syndrome. This review aims to describe the evidence for the use of this diet and how to deliver it in paediatric practice. A literature review covering studies on the FODMAP diet in adult and paediatric settings was conducted. While the evidence for the efficacy and safety of a FODMAP diet delivered in three phases, restriction, rechallenge and personalisation, is considerable, there is a lack of good-quality clinical trials exploring the efficacy of the diet in children and adolescents. Likewise, there are limited data on safety concerns associated with a restrictive diet in paediatrics, including impacts on nutrition and growth, disordered eating behaviours, psychosocial and family issues and families and the gut microbiome. The evidence suggests that the implementation of a dietary program is enhanced by a skilled dietitian when navigating a young person (and family) through healthy eating strategies and/or FODMAP restrictions to ameliorate their symptoms. Since the FODMAP diet is being prescribed globally to children, a practical guide for clinicians used to optimise efficacy and safety is provided, including the less restrictive ‘FODMAP-gentle’ diet.

## 1. Introduction

Since its beginnings, the primary focus of the FODMAP (Fermentable Oligo-, Di- and Mono-saccharides And Polyols) dietary strategy has been to improve gastrointestinal symptoms in individuals with irritable bowel syndrome (IBS). The majority of clinical studies are in adult populations. The prevalence of functional abdominal pain disorders in children is 13.5% worldwide, where IBS is reported most frequently, and self-perceived food intolerances negatively impact children’s quality of life [1,2]. Thus, it might be anticipated that the FODMAP diet should also be applied in paediatric populations. However, applying a restrictive diet in children and adolescents is not just about efficacy. Other considerations include nutritional adequacy and growth and adherence, as well as the social and psychological effects of a restrictive diet. Hence, the aim of this review is to outline general issues regarding the nature of the FODMAP diet and to explore its efficacy and safety in a paediatric setting, and how clinicians may apply this dietary strategy to reduce gastrointestinal symptoms in children.

## 2. Overview of the Fodmap Dietary Strategy

FODMAPs are dietary short-chain carbohydrates delivered to the large bowel due to: (a) slow absorption—fructose in excess of glucose and polyols, such as mannitol and sorbitol; (b) non-digestibility in the small intestine, particularly fructo-oligosaccharides (fructans) and galacto-oligosaccharides, due to the absence of suitable hydrolases in the intestine; or (c) reduced activity of relevant brush border hydrolases that might be genetically determined or acquired, lactose (hypolactasia) and, less commonly (and more controversially), possibly sucrose (hypomorphic forms of sucrase-isomaltase). The distribution of FODMAPs in the food supply is wide and outlined in detail elsewhere [3,4,5,6]. Their content in specific foods has been methodically mapped and the findings, expressed as low, borderline or high in specific FODMAPs per standard portion size, is readily available in the Monash University FODMAP Diet App [7].

The concept behind the FODMAP diet is that, because different FODMAPs have additive effects on the intestines, reducing the intake of all FODMAPs collectively in the individual will maximise the symptomatic benefit of such a dietary change on the symptoms. If such broad restriction has an adequate symptomatic response, individual FODMAPs may be reintroduced in a methodical way to identify specific dietary triggers and doses that are tolerated in the individual, followed by an adapted or personalised FODMAP diet that is nutritionally adequate and sustainable long-term [8,9]. This three-step, ‘top-down’ process is now referred to as ‘the FODMAP diet’ or ‘the FODMAP dietary strategy’ since a strict reduction in all FODMAPs (‘low FODMAP’) only occurs in the initial phase. On first principles, this process should be delivered by a FODMAP-trained dietitian, a view that is supported by one retrospective study [10], one randomised controlled trial [11], expert opinion and clinical guidelines [12,13,14,15].

The pivotal randomised control trial that provided proof of the efficacy of the first step in this diet was performed by our group in Australia using a strategy in which all food was provided in a controlled cross-over study [16]. The efficacy of this step in the diet has now been replicated on a global scale [16,17]. While randomised and controlled studies are not possible in defining the longer-term effectiveness of the personalised phase of the diet, prospective observational studies have provided convincing evidence of generally durable benefits with a minimal overall reduction in FODMAP intake in the majority [18,19,20]. While the application of strict research quality criteria (GRADE criteria) developed for assessing pharmacological trials rates the efficacy of the low FODMAP diet as ‘low-quality evidence’, it is recognised that dietary therapies seldom meet all criteria for high-quality evidence on the scale developed for assessing pharmacological trials [21]. Even so, the use of the diet has been included in many management guidelines as a primary or secondary therapy for IBS [14,15,22].

There have been several studies of the mechanisms underlying the effects of reducing FODMAPs [5,23,24,25]. The acute effects of ingesting FODMAPs are to distend the intestinal lumen via, first, their osmotic activity in the small intestine (most prominent for mono- and di-saccharides) moving water into the lumen of the small bowel, and, secondly, via excessive gas production from their rapid colonic fermentation in the proximal colon. The consequent distension of the intestinal lumen stimulates mechanoreceptors in the intestinal wall, which may lead to symptoms of bloating, abdominal pain and distension, and, secondarily, to changes in bowel habits [23,24]. The reason why there are differential responses to a similar FODMAP load in patients with IBS compared with those in healthy subjects is the presence of visceral hypersensitivity in the former [26]. The relatively rapid benefits of FODMAP restrictions over the first 7 days in patients with IBS are highly likely to result in a reduction in such distending events [27].

The longer-term effects of the FODMAP diet seems to maintain an adequate control of symptoms with minimal overall reduction of FODMAP intake [18,19,20]. Whether this is due to titrating FODMAPs to the correct dose of treatment and/or by correcting pathogenic mechanisms underlying visceral hypersensitivity is unknown. Potential mechanisms include: (a) induction of mast-cell activation by bacterial lipopolysaccharide [28]; (b) reversible injury of the colonic epithelium associated with increased intestinal permeability, mucosal inflammation and visceral hypersensitivity related to advance glycation end-products [29]; and (c) the durable correction of structural and functional dysbiosis of colonic microbiota in a proportion of patients [30]. Many of these mechanisms involve the intake of high amounts of FODMAPs and suggest that avoidance of high intake might permit correction of the resulting visceral hypersensitivity, as hypothesized [31]. In other words, restriction of FODMAPs might be more than just a symptomatic therapy and might correct some pathogenic events that drive IBS [30]. While further work is required to establish such a possibility, these ideas plus the longer-term clinical observations in adults are pointing to improvement in the natural history of IBS with the FODMAP diet. If this applies to paediatric populations with IBS, then there is even more imperative to define the safety and efficacy in younger patients.

## 3. Efficacy of the Fodmap Diet in Paediatric IBS

Currently, there is a lack of good quality clinical trials exploring the efficacy of the low FODMAP diet in children and adolescents. Three randomised control trials, one prospective observational study and one retrospective evaluation, all from different countries, as detailed in Table 1 [27,32,33,34,35], suffered from major limitations of trial design and/or reporting that included underpowering, a short duration of dietary intervention, poor assessment of dietary compliance and actual FODMAP intake, and for some, overinterpretation of pilot data. The designs and targeted populations are too heterogenous to be subjected to meta-analysis, but authors systematically reviewing the data have consistently concluded that efficacy of the low FODMAP diet in children is uncertain, but there are sufficient signals to warrant further studies of better quality [36,37,38]. A recent position statement by ESPGHAN concluded that there is scarce evidence to support the use of a low-FODMAP diet in children with IBS [39].

One major issue across all trials was that FODMAP composition data of habitual, prescribed or consumed diet was lacking or inaccurate. Although in practice it may not be practical or necessary to calculate FODMAP intake, in a research setting, this evidence becomes paramount in informing how restrictive of a dietary approach the clinician needs to take. In the first double-blinded randomised cross-over controlled trial in the USA, the FODMAP dietary prescription was defined as 0.15 g/kg/day or max 9 g/day, but total oligosaccharide was not reported and, therefore, whether their FODMAP prescription was achieved is unknown [27]. In the Turkish-based randomised control trial, a 0.5 g FODMAP per meal cut-off value was used, which was based on research from clinical trials in adults. However again, there were no available FODMAP composition data [34]. Other trials used more qualitative measures, such as dietary advice based on previously published food lists, but changes in FODMAP ingestion from habitual diet is unknown. Without precise assessment of the prescribed diet and dietary adherence, interpretation of degree of restriction necessary in children is unclear. 

In high-quality dietary intervention studies, measuring the actual intake of the targeted component is very important, as a measure of dietary adherence to the prescribed intervention, and of the ability of the educators to teach the diet, a factor seldom considered where the dietary intervention is being coached. The FODMAP diet is no exception. There is no current gold standard for measuring dietary adherence to a FODMAP diet. However, expert opinion recommends measuring actual intake of the dietary component in question [40]. This can be done utilising the Monash University database of FODMAP content of foods. Even though this was originally based upon foods sourced in Australia, the FODMAP content of primary foods has largely been validated when internationally-sourced [3,6,41]. Therefore, because no current paediatric studies have complete FODMAP composition data, there has been a reliance on arbitrary measures. For example, a USA-based study investigating the nutritional adequacy of the low FODMAP diet used a rather unique method to determine adherence [42]. First, the average number of high FODMAP foods consumed per day at baseline was calculated, and then an arbitrary cut-off value was applied to this habitual intake of high FODMAP foods. The number of high FODMAP foods that was below intake for >95% of participants was deemed ‘adherent,’ which resulted in 3 high FODMAP foods. Thus, participants could eat up to 3 high FODMAP foods per day to be considered adherent to the low FODMAP diet. However, without actual FODMAP intake data this method presents issues. Whether this approach of using baseline diets can be validly used as a benchmark for adherence is debatable, because whether the outcome of the dietary intervention actually results in a ‘low FODMAP’ diet is unknown. Adherence has potential to be higher in feeding trials where all or most foods are provided to participants—both the Polish and USA based feeding trials asked participants to return leftover uneaten food packages, and document any additional foods that were not provided [27,35]. Adherence was defined as a percentage of amount of food eaten, or number of times non-compliant per week in the US and Polish studies, respectively. Despite there being limitations in these arbitrary evaluations, they are in line with studies in adults [16,43], and at least provide some insight, where some paediatric studies make no mention of adherence at all.

Choice of control is challenging in dietary trials and dependent on the dietary component in question. The ‘typical American childhood’ diet was the control used in the US-based study. Both diets were matched for nutrient composition and, where possible, an identical food that differed only in FODMAP content was used across both diets [27]. Given the wide variety in diets, it is difficult to determine a ‘typical’ standardised diet to use, but this choice of control is in line with the our original RCT in adults, that used a ‘typical Australian diet,’ as the control diet based on Australian healthy eating guidelines [16,27]. Albeit, there were limitations in duration of the intervention in the paediatric study lasting only 48 h, which may not have been enough time to see the full effect of the diet, particularly as it took up to 7 days for benefits in the adult feeding study [16]. NICE guidelines were used as control in some adult and one paediatric study [35,44,45,46]. The NICE approach is problematic as a comparator for two reasons. First, the guidelines, while mainly about the style of eating, advise restricting fruit intake, avoiding certain high-fibre foods such as wholemeal flour and breads, and avoiding sorbitol from sugar-free products. These recommendations would reduce intake of fructans, excess fructose and polyols. Secondly, the efficacy of the NICE Guidelines diet has never been subjected to evaluation against placebo. A network analysis of dietary interventions in adults with IBS indicated it may have efficacy itself (though less than the low FODMAP diet) [47]. When the increment of benefit from the comparator is uncertain, it is very difficult to power a study of another intervention against it, as was evident in a very recent report and its associated criticism [45,48]. The researchers from the Polish comparative study suggested that the NICE diet was better than specifically reducing FODMAPs, but the number of children studied was very small and actual FODMAP intake and its change in the two groups were not measured. Making conclusions from such pilot studies has to be regarded with great scepticism, but should be used to inform properly-powered studies. An example of this approach was the suggestion (borderline statistical significance) in adults with self-reported gluten sensitivity that gluten was the culprit [49]. When a well-powered study was subsequently performed by the same group, no effect of gluten rechallenge was observed [50]. 

In the design of dietary intervention studies in paediatric patients with IBS, the choice of endpoints is important. However, this aspect is not as controlled by authorities, such as the FDA as in adults. The way response was assessed in the published paediatric studies is heterogeneous and renders meta-analysis, for example, challenging. Clearly, the endpoints need to involve both abdominal symptoms, quality of life and behavioural measures. 

## 4. Safety of the Fodmap Diet in Paediatrics

Any dietary strategy that involves dietary restriction faces specific questions regarding safety. Those issues that particularly apply to children and adolescents are discussed as follows.

### 4.1. Nutrition and Growth

In adult studies, questions have been raised over the nutritional adequacy and diet quality of the low FODMAP diet [42,51]. For example, calcium intake may be lower given the restriction of some dairy products, with mixed findings from clinical trials in adults [20,51,52,53,54]. After adjusting for energy in a study comparing NICE guidelines with a low FODMAP diet in adults, there was a statistically significant decrease in riboflavin and an increase in niacin and vitamin B6 in the low FODMAP group, and no decreases in the NICE group after 4 weeks [53]. However, there was no significant change in the proportion of participants meeting dietary reference intakes for riboflavin, niacin or vitamin B6. In a New Zealand-based study of community-dwelling older adults, there were no significant differences after 6 weeks following the diet in macro and micronutrients [54]. In addition to dietary advice, these participants were given food lists, recipes, meal plans, shopping guide and information sheets, which clearly supported the participants in making dietary changes. It would be anticipated that, in general, greater support from a dietitian in the community would be associated with improved nutritional intake compared with advice online or just from food lists. When interpreting results commenting on the nutritional adequacy of the FODMAP dietary approach, method of delivery must be considered. 

There seems little doubt that examining single nutrients fails to capture the complexity of the diet, and that we should instead be focusing on diet quality, relating to how closely the diet aligns with dietary guidelines [51]. In a secondary analysis of two randomised trials, 4 weeks on a dietitian-delivered low FODMAP diet in 63 adults with IBS, there were no significant differences in the intake of most nutrients compared with that of 67 patients receiving a sham or habitual diet. However, when then assessed broadly (for example considering variety of food groups and nutrients), quality of the low FODMAP diet was inferior to that of the habitual diet [51]. This does not necessarily mean the diet promotes unhealthy eating, but indicates the potential effects even under expert dietary guidance. 

The prevention of nutritional compromise when instituting a low FODMAP diet is of even greater importance in children as deficiencies in macro and micronutrients may impact adequate growth and development. While clinical trials generally involve a specialist gastroenterology dietitian, access to a dietitian who is skilled in both paediatrics and the low FODMAP diet may be limited in practice. A 2020 retrospective chart review of children with functional gut symptoms found that, of the 115 children provided dietary advice, only 23 (20%) received this from a registered dietitian [55]. This is an alarmingly low rate of dietetic involvement, given the potential risks of placing children on restrictive diets. 

While there are limited data examining the nutritional adequacy of a FODMAP diet in children, there is some evidence to suggest that instituting a low FODMAP diet under the guidance of a dietitian may in fact improve diet quality. For instance, a U.S. study of children aged 7–13 years with functional abdominal pain disorders, showed their habitual diet was of poor quality [42]. The children were not meeting recommended servings for fruits, vegetables, grains or dairy, and average intakes of various vitamins and minerals (vitamins A, C, D, E, as well as calcium, phosphorous, magnesium, potassium, manganese and choline) were all below recommended daily allowances. Results like these are not unique to this study, poor quality diets in children are observed across the USA [56]. After the 3-week dietitian-led low FODMAP dietary intervention, children’s FODMAP intake decreased, and there was improved intake of vitamins B6, C and E, and serves of vegetables and protein (*p* < 0.05), while intake of refined grains, sweets, ultra-processed foods, trans-fatty acids and added sugars decreased [42]. 

The two principal reasons for the improved nutritional intake are first, the habitual diet was very poor. Secondly, the U.S. study involved 3 intensive 1-h sessions across 3 weeks from a specialist dietitian, including extensive educational materials (food charts, menus, access to a smartphone app, weekly homework). These multifaceted educational strategies are an exemplar for FODMAP education for families, albeit perhaps not as accessible for those in the community, and help to explain the high adherence rates and improvements in nutritional quality. Hence, the enhanced dietary quality was not specific to the low FODMAP diet itself, but to education on making better food choices and eating behaviours. Further studies are needed over a longer duration to support this. Whether dietary restrictions from the FODMAP diet impact on growth during different developmental stages across childhood or adolescence is yet to be determined. 

### 4.2. Disordered Eating

The vast majority of patients with IBS and other DGBI recognise food ingestion is associated with symptom genesis. Hence, many patients adjust their eating patterns in an attempt to ameliorate their symptoms. In other words, ‘disordered’ eating patterns are almost appropriately common in patient with IBS. On this background, identifying those with truly disordered eating behaviours or even eating disorders can be challenging. This is, however, important since introducing a restrictive diet is not considered appropriate in patients with pathological eating patterns. Unfortunately, screening tests, like the SCOFF (Sick, Control, One, Fat, Food) instrument [57], have not been validated in patients with DGBI, in that these instruments do not cater for food avoidance related to symptom management. For example, the question, ‘would you say food dominates your life?’, may not necessarily be related to psychiatric issues but more to their IBS symptoms [58,59]. Such limitations underlie the difficulty in defining prevalence of psychiatrically relevant disordered eating behaviours or formal eating disorders such as anorexia nervosa, bulimia, ARFID (avoidant/restrictive food intake disorder) or orthorexia (obsession and fixation on self-perceived healthy eating) [60,61]. 

Prevalence can be examined from two directions. First, the frequency and nature of disorders of DGBI can be examined in an eating disorders outpatient clinic. In the USA, 72% of participants reported at least one bothersome gastrointestinal symptom, with 39% meeting criteria for a DGBI with functional dyspepsia being the most frequent pattern [62]. The study included individuals between the ages of 8 and 76 years, but did not distinguish frequency between children and adults. 

Secondly, disordered eating behaviours have been studied in populations with DGBI. According to responses to the SCOFF, 23% of 233 adults patients with IBS were at-risk of disordered eating behaviours [59]. In a multidimensional analysis of 955 adult patients with IBS, 13% reported severe food avoidance and restriction with strong associations with reduced quality of life and heightened psychological symptoms [63]. A recent systematic review stated that the prevalence of disordered eating in adults with gastrointestinal disorders ranged from 13–55% and that this was higher in those with DGBI compared with those with organic gastrointestinal disease [64]. 

Data are even more imprecise in the paediatric population. There are few doubts that eating-associated symptoms are common. In a study of adolescents with IBS, compared with healthy controls, eating-associated symptoms were more frequent (91% vs. 28%, respectively; *p* < 0.001) and severe in the IBS group [65]. Some of the strategies the children use to control their symptoms are worrying. They included, not eating any food even when hungry (42% vs. 16%; *p* < 0.001) or vomiting after eating (13% vs. 2%; *p* < 0.05) [65]. This is of great concern because children and adolescents are in the process of developing and establishing their relationship with food and eating patterns, and, during teenage years, they are more likely to manifest eating disorders and have body image issues [66,67,68]. However, the prevalence of severely disordered eating patterns in paediatric populations attending gastroenterology clinics appears much less than in adults. For example, in a retrospective study of 2231 consecutive referrals in Boston, 1.5% fulfilled criteria for (ARFID) and a further 2.4% met one or more ARFID criteria [69] in contrast to 10–20% reported in adult populations [70]. Whether the same is true for patients with DGBI has not been determined.

The concern of many health professionals, particularly psychologists and dietitians, has been that prescribing the FODMAP strategy will exacerbate or precipitate an eating disorder. The risk and safety of such a scenario has yet to be well described. The only relevant published data come from the 233 adult patients with IBS in whom 23% were at-risk of disordered eating behaviours according to the SCOFF prior to dietary instruction [59]. This subset showed greater adherence to the diet, but the SCOFF after dietary implementation did not identify a greater proportion of at-risk patients [59]. There are no data in the paediatric population. Nevertheless, it must be considered that children and adolescents with disordered eating behaviours, which might include fussy eating, skipping meals, food aversions and/or body image issues, may compromise the nutritional adequacy of an already limited diet and exacerbate disordered eating patterns or develop psychiatric illness with further dietary restriction via the avoidance of high FODMAP foods. In the context of fixations or obsessions on dietary changes for overlapping IBS and eating disorders in children with IBS, non-restrictive diet therapies (e.g., fibre supplementation), or non-diet therapies (e.g., psychological approaches, medications and herbal remedies) may be more appropriate in these contexts, but require involvement of a multidisciplinary team [71,72,73]. Also, tools to accurately identify children who are at risk of disordered eating and/or suffering eating disorders are needed for the DGBI population. 

### 4.3. Psychosocial Impacts on Family and Adherence to the Diet

The nature of restrictive diets in a paediatric population means that the entire family unit is impacted, where parents play a key role in influencing the dietary habits of their children. Currently, there is minimal good quality evidence investigating adherence to the low FODMAP diet and impact on quality of life for children and their families. 

In the study comparing low FODMAP to NICE diet, children and parents had similar responses regarding abdominal pain intensity and frequency, with a high compatibility between answers (85% FODMAP, 93% NICE) and frequency (77% FODMAP, 86% NICE). However, a low compatibility regarding stool consistency (23% FODMAP, 43% NICE) [74]. Despite not formally evaluating the reasons as to why there was a large difference in compatibility of answers, the authors concluded that this may have come down to children’s embarrassment about talking to their parents about defecation [74]. Bowel habits are a sensitive and personal, yet crucial topic related to IBS research, and if children do not feel comfortable to speak to their parents, they may not have access to appropriate healthcare to improve their condition. Also, there are no current studies investigating the impacts of the FODMAP diet on the quality of life of children and their families. The diet improves quality of life in adults, but this must be also replicated for children [75,76,77]. Future studies should address these barriers and provide strategies to ensure that children feel comfortable to report on the sensitive topic of defecation. 

Parental opinion about comprehension of the dietary recommendations was assessed as part of the study comparing the efficacy of the low FODMAP and NICE diets in children with functional abdominal pain [74]. A significantly lower percentage of parents in the low FODMAP group found the diet easy to follow (38% vs. 57%), ‘ease’ being defined as product availability, acceptability by children and difference from usual diet. Families were blinded to the intervention and randomly allocated to receive a diet of prepared meals for their child. There was no difference in understanding the written information or satisfaction with the improvement in child’s symptoms [74]. Compliance, measured by the number of additional ‘not allowed foods’ eaten per week that were not part of the provided food, was significantly different in the second week of the intervention, with a median of 6 times per week in the low FODMAP group, compared with 1.5 in the NICE group [35].

Unlike the study above, in a New Zealand based study, patients (aged 4–17 years) and their families were provided with formal dietary advice across all three phases of the FODMAP diet. Two out of 3 found the written information easy to follow, 79% found the diet easy to follow, 69% were satisfied with their improvement in symptoms and 59% were not interested in changing their diet further for symptom reduction [33]. Such good results may have reflected the nature of the educative process and written information provided. This study also highlights another issue, commonly seen in paediatric trials, where a broad range of age groups are studied together. The physiological and psychosocial needs within each group are dynamic and unique as one moves through childhood into adolescence. This is reflected in the same cohort where better compliance to the low FODMAP diet was in children under the age of 10 (56%) compared to those over the age of 10 (45%) [33]. Whether this could be explained by a higher degree of parental involvement was not explored. Thus, it would be useful if future studies isolate age groups, for example, 5–12 years old and 13–17 years old, so that intra and inter group comparisons can be made—this may identify age-specific issues and needs of children and adolescents presenting with IBS.

The potential for major impacts on activities, daily life and participation in the community for children and their families of teaching a restrictive diet in children were exemplified by a study of gluten-free diet for coeliac disease [78]. Positive impacts include learning new culinary skills, while negative ones include increased expenses, increased time spent coordinating with institutions as well as limiting choices in restaurants [79]. There are, however, major difference between gluten-free and FODMAP diets. First, the gluten-free diet is a strict lifelong diet for coeliac disease in contrast to strict FODMAP restriction, which is intended for the short-term (2–6 weeks), before being reintroduced to identify FODMAPs. Secondly, the restriction is absolute for the gluten-free diet and only reduced for the FODMAP diet. Thirdly, the implications of straying from the diet are different with gluten intake driving the disease whether it induces symptoms or not as opposed to transient symptoms with increased FODMAP intake and the ability to adjust the level of restriction according to symptom burden. Fourthly, the low FODMAP diet is more complicated as FODMAPs are found across the entire food supply and are not subject to the same food labelling requirements as those following a gluten-free diet. Lastly, community exposure and awareness for gluten-free diets are better than for FODMAP diets, so gluten-free catering at food outlets is often available, at least in many countries [80,81,82]. 

The low FODMAP diet in children will have unique characteristics that affect the child and family. There is a need to study this further and, for now, clinicians need to ensure that the education and resources they provide to families are clear and well-understood. Table 2 provides strategies clinicians can use to keep parents engaged during their consult regarding the FODMAP diet. Food lists, smartphone applications, recipes, education regarding label reading, information regarding global food certification programmes and regular follow up are all strategies that can be employed by the dietitian to support the family and assist with adherence to the diet [9,83]. 

### 4.4. Effect on Gut Microbiota

The concern with regards to the effects on the gut microbiota are that carbohydrates have a powerful influence on the composition and fermentative activity of the gut microbiota [84]. The long-term effects of such changes on gut and general health, and the concept that many FODMAPs have prebiotic actions have created concerns. Fortunately, in adults, apart from a lower absolute abundances of Bifidobacteria spp., there are no clear effects on microbial diversity or abundances of specific taxa [84]. There is also now evidence in adults to suggest that the low FODMAP diet may indeed shift a subset of individuals with IBS to a ‘healthier’ microbiome [30], challenging the idea that FODMAP restriction is detrimental, at least in the short-term. Adverse health outcomes have yet to manifest and the personalised phase of the diet has been associated with few changes in the microbial community [19]. There are no data on the effect on faecal microbiota of restricting FODMAPs in paediatric cohorts, but it is likely that these would mimic those in adults. 

In an era of personalised medicine, there is considerable interest in predicting which patients will experience symptomatic improvement on a low FODMAP diet. Community bacterial structures in the faeces predictive of the response to reducing the FODMAP intake have been sort. In adults, certain patterns of faecal microbiota and volatile organic compounds have predictive value of varying strength for response to the Phase 1 of the diet [85,86,87,88]. In the paediatric population, 16S rRNA gene amplification of faecal bacteria demonstrated enrichment in several OTUs with, for example, greater saccharolytic metabolic activity prior to commencing the diet in children with IBS who had ≥50% decrease in abdominal pain episodes within 48 h of instituting a low FODMAP diet [27,89]. The clinical value of predicting response to reducing FODMAP intake via faecal microbiota assessment, however, remains uncertain without prospective studies.

## 5. Practical Delivery of the Fodmap Diet

Optimal implementation of the FODMAP diet in the adult has been well described [90]. The perspectives applying to its introduction to children are outlined below. 

### 5.1. Role of the Dietitian—What They Do

In order to mitigate the risks associated with introducing a restrictive diet to paediatric patients with IBS and to achieve optimal clinical outcomes, assessment and coaching by a FODMAP-trained dietitian is essential. While this view cannot be backed up by strong evidence, adults who seek guidance from a dietitian are more likely to succeed in adherence to the diet and effectively reduce their gastrointestinal symptoms [10]. Interestingly, in a Belgium based study where GI physicians spent 20–30 min with IBS patients (5–10 min spent discussing FODMAPs) and provided only a leaflet for the 6-week intervention, at the end of the trial, 60% of patients stated they would have liked to have seen a dietitian before starting the diet [91]. Furthermore, dietitian-led education is superior in clinical response to brief education with food lists as often applied in clinical practice [11]. FODMAPs are found in core food groups, including fruits, vegetables, grains, legumes, dairy products and nuts in addition to discretionary food items. As children are establishing food habits, careful considerations and appropriate substitutions must be made if choosing to reduce FODMAP intake from core foods.

There are several general goals that the dietitian should be seeking to fulfil during the consultative process, with individual tailoring. Table 3 details behavioural and practical considerations that must be considered during the initial consult with a paediatric patient with gastrointestinal symptoms according to age group.

**Goal 1**. Evaluate the diagnosis: For children presenting with or without referral from a medical practitioner, it behoves the dietitian to be comfortable with the diagnosis and to ensure adequate investigation, such as coeliac serology, has occurred. Children presenting with diagnosed or undiagnosed DGBIs may previously have seen alternative medicine practitioners (e.g., naturopaths) who offer patients a range investigations, pseudo-diagnoses and treatments, many of which are devoid of scientific principle. Pseudo-diagnoses to look out for in paediatric patients include fructose malabsorption or intolerance, bacterial dysbiosis, non-coeliac gluten/wheat sensitivity (this may be a legitimate diagnosis, but often is not) and food ‘allergy’ or intolerance (these may or may not be legitimate diagnoses, depending on who made it and how it was made). Differential diagnoses should also be considered in children, such as constipation with overflow diarrhoea, which can mimic the symptoms of diarrhoea-predominant IBS and non IgE-mediated allergy. 

Screening for ‘red flags’ or ‘alarm features’ is important in all children presenting with a diagnosed or undiagnosed DGBI. If identified, these indicate the possible presence of conditions other than IBS and require further medical investigation or treatment if not already addressed by the child’s doctor. Red flags and the conditions they may indicate are summarised in Table 4.

**Goal 2**. Assess the nutritional quality of the habitual diet: This is an important first step in which nutritional deficiencies and excesses will be documented and eating patterns defined. Such dietary evaluation is important for general health, particularly given the rising prevalence of childhood obesity in most Western countries, driven largely by excessive intakes of discretionary food items such as sweetened beverages and highly processed foods [92,93]. FODMAPs are found across the whole food supply and, outside of a few foods that are particularly concentrated in certain FODMAPs, portion size ultimately determines whether most foods are deemed to be ‘low’ or ‘high’ FODMAP according to established cut-off values [94]. Examples of higher FODMAP, less nutritious foods are sweetened beverages and confectionery (containing high-fructose corn syrup in some countries), and potentially other packaged food items that may be designed and marketed to children. High-FODMAP foods of greater nutritional quality that are more commonly consumed by children include some fruits, wheat products and lactose-containing milks (relevant if the child has hypolactasia).

Fruit juices containing high doses of excess fructose have been implicated childhood diarrhoea, where early studies found simple removal from the diet was enough to alleviate symptoms in toddlers [95]. Children under the age of 10 have a reduced capacity to absorb fructose [96], and the dietitian should pay careful attention to fluids in their dietary assessment. Interestingly in a study evaluating FODMAP content of children’s food products in USA, even foods that were selected on the basis of potentially being low in FODMAPs, such as gluten free bakery items and breakfast cereals, were laboratory tested and contained excess fructose [97]. 

In younger children, it is important to assess eating/feeding skills and behaviours to identify whether feeding difficulties exist. Feeding difficulties, such as fussy eating and food refusal, may limit the variety and volume of food intake, as well as the child’s acceptance of new foods. Therefore, it is important not only ask about what foods are consumed, but also what foods are accepted. Collecting this information for each food group will help to identify nutritional gaps; to come up with suitable low FODMAP alternatives, and to provide tailored dietary advice. A low FODMAP diet may be contraindicated if the child will not accept substitute, low FODMAP foods. 

**Goal 3**. Assess eating behaviour: Eating behaviour may look different for different age groups. For example, as children enter their teenage years, behavioural changes include skipping meals, eating foods away from the home and increased high energy, low nutrient snacks occur more frequently [98]. Therefore, establishing regular meal patterns and general healthy eating habits are warranted in initial consultations. Consequently, spreading intake across the day may also spread FODMAP intake more evenly, but the focus is around healthy eating, rather than FODMAPs. Conversely, skipping meals may induce or exacerbate symptoms by, for example, promoting constipation through reduce colonic motility as suggested by a lack of gastro-colic reflex [99] or by consumption of greater amount of food at one meal. The dietitian should assess for disordered eating habits or an eating disorder where benefits from restricting the diet further are unlikely or contraindicated.

**Goal 4**. Document nutritional status: Simple anthropometry and plotting on growth charts is essential to be documented for both identification of nutritional problem (such as undernutrition or overnutrition) and for follow up. Importantly, sub-optimal growth, namely weight loss, low weight for age (underweight), low weight for height (wasting), and low height for age (stunting) are atypical features in IBS and warrant further investigation.

**Goal 5**. Correct the overall nutritional adequacy of the diet: Assessing current FODMAP intake is important to identify possible symptom triggers and to determine the level of FODMAP restriction required. The child’s FODMAP intake should be considered in reference to nutrition guidelines regarding the number of serves of food that should be included from each of the five food groups. For example, if a child has a very high FODMAP intake on daily basis due to excessive intake of fruit/fruit juice and/or grains, FODMAP restriction may not be necessary. Rather, the child may achieve adequate symptom relief with dietary advice to consume a balanced diet that limits fruit and/or grain intake to the recommended daily serves.

In some children, the initial focus may be to correct the overall dietary structure by removing energy-dense foods and swapping them with healthier options, rather than focusing on education of FODMAP reduction. Such a strategy may be implemented not only for promotion of healthy eating for nutrition and disease prevention, but for its therapeutic potential. The above example where removal of fruit juice resolved symptoms was in a population of toddlers, thus, in older children with an increased capacity to absorb fructose, the dietitian may instead choose to reduce portion size of juice to be in line with dietary guidelines. It may also be expected that a poor-quality diet that is deficient in fibre may contribute to gut symptoms such as constipation; it may be prudent to correct such poor intake in the first instance. This strategy is backed by evidence. First, in an uncontrolled, unblinded Spanish study of 21 children and adolescents with IBS, baseline diet was characterised by a high intake of refined cereals, sugary drinks, sweets and pastries [100]. Oral and written dietary advice with a focus on healthy eating led to improvements in the children’s overall dietary habits and weight status [100], but importantly, also improved gastrointestinal symptoms of abdominal pain, bloating, gas, constipation and diarrhoea. Secondly, as mentioned, a US-based study found poor dietary quality at baseline assessments, with improvements in quality after implementation of the first phase of the FODMAP diet [42]. Therefore, correcting nutritional inadequacy is at the discretion of the dietitian, whether they choose to this by reducing FODMAPs or through general healthy eating strategies. 

**Goal 6**. Decide what FODMAP intake to target: There are two potential approaches in reducing FODMAP intake. The first is to aim for a specific intake of FODMAPs, which has been made possible by detailed information on FODMAP content in foods. This approach has been taken in several adult research studies. As food composition lists with FODMAP content have evolved, arbitrary cut-off values for FODMAP content in meals that will effectively consider food and meals as a whole as ‘low FODMAP’ have been developed [94]. A conservative figure of 0.5 g FODMAPs per main meal have been used in feeding trials of adults, with extra snacks accumulating to up to approximately 3 g/d, which has proven to be successful in ameliorating gastrointestinal symptoms in the majority of adults with IBS [16,101]. Importantly, in studies in adults where the low FODMAP intervention is provided by dietary advice, rather than provision of foods, adequate symptom reduction has been observed where deviations from the strict 0.5 g FODMAPs per meal are more likely [43]. 

The second approach, which is more amenable to routine clinical practice, is to reduce the FODMAP intake without a specific final FODMAP content target. Knowledge of habitual intake of FODMAPs is a key to this approach and underscores the importance of incorporating such an assessment outlined in Goal #2. The impact of pre-intervention FODMAP intake was evident in a randomised controlled trial in which a relatively low proportion of adults with IBS responded to a low FODMAP diet [44]. In that study, the magnitude of difference in FODMAP intake quantified via food diaries between habitual and intervention diets was small (after lactose intake was controlled for which was reasonable since the prevalence of hypolactasia was likely very low in the study population).

In paediatric studies, measurements of actual habitual or interventional intake of FODMAPs have not been performed in any of the reported studies and, while an aspirational target intake was arbitrarily defined in the U.S. study, whether this was achieved is not known since oligosaccharide content and total FODMAP composition was not reported. In a formal re-challenge study in which symptoms were provoked by specific FODMAPs, a very large dose (19 g) of fructans in a water solution, in contrast to maltodextrin as placebo, consumed across three meals showed fructans to be potential triggers of symptom [102], but provided no dose-related information. Experience in adult clinical practice is that there is a wide variation of dose and types of FODMAPs that provoke symptoms in the re-challenge phase [33,103], as was reported in the New Zealand experience in children [33,103]. 

Since satisfactory amelioration of gut symptoms have been reported in the majority of children without quantitative intake data, the application of strict conservative cut-off values for total FODMAP intake may in general not be needed. A simplified version of the diet could be more appropriate in this context. Such considerations led to the ‘FODMAP gentle’ concept [104] as illustrated in Figure 1. The general principle is that only the foods most concentrated in FODMAPS across the food supply, such as garlic, onion, wheat-based products, apple, pear, stone fruits, mushroom, cauliflower, lactose-containing milk and yoghurt, be restricted. Although this approach has not been investigated, theoretically, the benefits of a less restrictive yet efficacious diet are appealing in the paediatric setting. As mentioned, dietitians may initially correct nutritional inadequacies or use the FODMAP diet to promote more healthful eating; the FODMAP gentle concept accommodates both scenarios. It is a less restrictive diet, where only some foods are replaced with low FODMAP alternatives. Because it is less complicated, it becomes more appropriate for certain demographics such as children and families with lower health literacy, patients with other dietary restrictions, bigger families with other competing priorities and other scenarios where nutritional adequacy is challenged by fussy eating or associated disease (e.g., diabetes, inflammatory bowel disease).

In order to institute the ‘gentle’ approach, it is important to identify and subsequently target the major high FODMAP foods that children eat. This is likely to be region-specific. Additionally, foods high in fructans and fructose are commonly seen in children’s diets, the majority of fructans coming from wheat-based foods, the majority of fructose coming from grains (including processed breads/cereals), fruits and fruit juices in studies from USA [4,105]. This is not be surprising seeing as children are among the highest consumers of bread, breakfast cereals and fruit [106]. In a study of FODMAP intake in a Spanish cohort, the main FODMAPs in the diets of children and adolescents were lactose, excess fructose and fructans, with whole cow’s milk, apples, and white bread ranked highest [107]. In Spain, white bread and biscuits contain both fructans and excess fructose [107]

**Goal 7:** Consider long-term implications: It is up to the discretion and clinical reasoning of the dietitian regarding the long-term application of the diet, because currently, there is not enough information in regards to the third phase or personalisation, where higher FODMAP foods or portion sizes that are well tolerated remain part of the habitual diet. As we know, children’s food preferences and eating behaviours are heavily influenced by their environment, which changes dramatically over the course of childhood and adolescence. The dietitian may not have long-term contact with the family, particularly if the symptoms have resolved in a relatively short-term period. Thus, it is necessary for the dietitian to educate the child and family on the fluctuating nature of symptoms over time and the importance of long-term reintroduction of foods to assess for tolerance with aim of keeping the diet as minimally restrictive as possible. Future studies investigating long-term effects of the diet should also include information on growth, potentially through the use of growth charts, because for now, how variations of the FODMAP diet fit in the context of developmental phases is unknown. 

### 5.2. Future Directions in FODMAP Treatments

So far, research into the FODMAP diet has looked mainly into refining the therapy for IBS and other GI conditions, such as IBD, colic, endometriosis and functional dyspepsia, but now, we should consider how else the FODMAP diet can evolve [108,109,110,111,112]. Other dietary carbohydrates can potentially act as FODMAPs. Just as lactose is a FODMAP *only* when brush-border lactase is deficient, reduced activity of brush-border sucrase-isomaltase may lead to sucrose and starch oligosaccharides to be FODMAPs by virtue of some escaping hydrolysis. Additionally, the use of digestive enzyme preparations presents another potential treatment modality beyond lactase that is established in patients with hypolactasia.

### 5.3. Sucrase-Isomaltase Deficiency

Congenital SI deficiency is a rare condition that has long been recognised. However, hypomorphic forms of SI in the brush border of the small intestine, identified by genetic analysis, appear associated with adults with diarrhoea [113]. SI hydrolyses sucrose and oligosaccharides released from amylase digestion of starch. Conceptually, there is a wide range of abilities to complete such digestion and reduced activities of SI may lead to sucrose and starch be escaping digestion in sufficient quantities to induce symptoms and, in other words, to act as FODMAPs. In a study of 308 children presenting with idiopathic loose stools, compared with the general population, the cumulative prevalence of hypomorphic SI variances was higher than in the general population (4.5% vs. 1.3%, *p* < 0.01) [114]. In a smaller study of 31 adults with a diagnosis of IBS-D or IBS-M, reduced SI activity on duodenal biopsy was present in 35% of patients [115]. Interestingly, retrospective analysis of 46 adult patients with IBS-D who participated in a trial of a low FODMAP dietary, those that were carrying SI hypomorphic variants benefited from the diet significantly less compared to non-carriers (44% carriers vs. 61% non-carriers. *p* = 0.031). The application of a low-sucrose and starch diet appeared to have efficacy in uncontrolled observations in 74% of 80 in patients with IBS (any bowel habit) or other DGBI [116]. Whether the reduction of sucrose or starch was the reason for benefit is uncertain. 

Putting all of this together, there is a chance that a small percentage of children with abdominal symptoms and a diagnosis of IBS may indeed have reduced activities of SI as a contributor to their symptoms and require additional restriction of sucrose and/or starch in association with other FODMAPs. However, this is entirely speculative at this point in time. Indeed, such a strategy should be embarked upon with care as starches are a major and important component of a child’s diet as an energy source for growth and development. For researchers, SI deficiency should also be kept in mind when designing placebos. For example, the maltodextrin was used as the placebo when the effect of fructan challenge was being assessed in children who had responded to a low FODMAP diet [102]. Future studies are needed in this area as validation of the clinical tests for SI deficiency (breath tests, duodenal enzyme activities, genetic testing) could assist in understanding poorly responsive to diagnose unexplainable abdominal symptoms and lead to efficacious therapeutic approaches [117]. 

### 5.4. Supplementary Digestive Enzymes

Enzyme therapy is an attractive treatment option. If such enzymes have efficacy, they might reduce the burden of dietary intervention, something particularly appealing to busy families. The use of lactase supplementation, with food or within food, has an established role in reducing symptoms of lactose intolerance and reduce the risk for calcium deficiencies [118]. There is some evidence that supplementation with oral alpha-galactosidase, the enzyme responsible for digesting galactooligosaccharides, will improve symptoms in adults, but only in those sensitive to galactooligosaccharides [119]. However, it was ineffective in an unselected population of patients with IBS [120]. Several multi-enzyme preparations that target fructans and other FODMAPs are now marketed to reduce the FODMAP load in food and subsequent symptom generation. Unfortunately, evidence of efficacy is limited to anecdotes. Digestive enzyme supplements potentially represent the opportunity to create new treatment options for children with IBS, but studies demonstrating efficacy and safety are required before they can be endorsed. 

## 6. Conclusions

It is somewhat disconcerting that the FODMAP diet is widely used in paediatric practice with such weak evidence for efficacy. This reflects the facts that expert opinion and anecdotes have great influence in application of diet therapies and that high-quality clinical trials assessing diet therapies are so challenging to design and execute. Despite this relative lack of quality evidence for efficacy, the FODMAP diet is being prescribed globally in children and, therefore, considerations need to be made by healthcare professionals. Awareness of the risks of the diet, and their mitigation by adequate assessment and coaching by skilled specialist dietitian, are crucial to ensure optimal outcomes and tailoring to the needs of the child and family. Individualisation of the approach taken is important; some may not be suitable for the FODMAP diet, some may require initial correction of nutritional deficiencies with general healthy eating strategies before adopting the FODMAP diet, while others might be best served by using the FODMAP diet to assist in both correcting nutritional deficiencies and IBS symptoms. While the ‘FODMAP gentle’ concept has not been rigorously tested, the use of a simpler, less restrictive approach in a paediatric setting is practical and sensible, takes into consideration the broader needs of the family unit and should be considered by clinicians. There is an ongoing need for quality research in all these areas.

## Figures and Tables

**Figure 1 nutrients-14-04369-f001:**
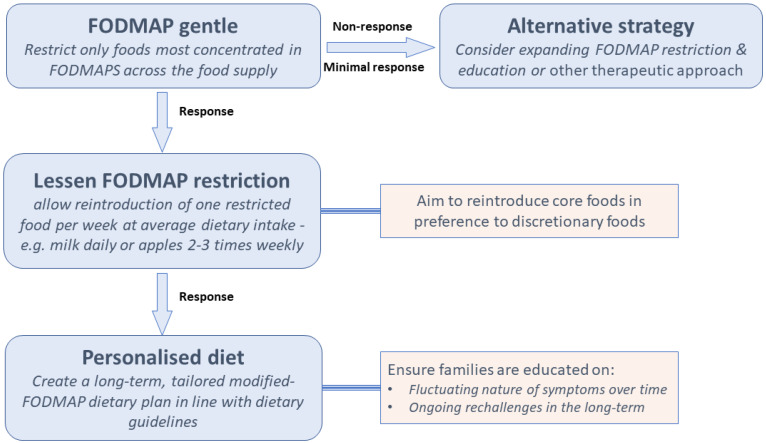
FODMAP gentle proposed framework as it applies to children.

**Table 1 nutrients-14-04369-t001:** Summary of trials evaluating the effect of a low FODMAP diet on gastrointestinal symptoms in children.

Study	Design	Patients	Intervention(s)	Endpoints		Comments
				Abdominal Symptoms	Other	
Chumpitazi 2015 USA [27]	Double blind randomised cross-over control trial	*n* = 33 7–17 years IBS (Rome III)	Low FODMAP diet: 0.15 g/kg/d FODMAPs or max 9 g/day vs. Typical American childhood diet (TACD): 0.7 g/kg/d FODMAPs or max 50 g/day) All food provided 48 h intervention 5-day washout in between diets	Abdominal pain: Fewer daily episodes during low FODMAP (1.1 ± 0.2 (SEM) vs. 1.7 ± 0.4 TACD, *p* < 0.05) Compared to baseline, there were fewer daily pain episodes (1.4 ±0.2) during the low FODMAP arm (*p* < 0.01) Responders (≥50% decrease in number of pain episodes) (*n* = 8), non-responders (*n* = 15), placebo-responders (*n* = 10)	Compliance based on food records was similar TACD (85%) and low FODMAP (91%) Responders to the low FODMAP diet enriched at baseline in several OTUs with greater saccharolytic metabolic capacity	Short intervention of only 48 h Study designed to examine predictive microbiota
Baranguán 2019 Spain [32]	Prospective observational	*n* = 20 5–15 years Functional abdominal pain	List of foods categorised as ‘allowed’ or ‘not allowed’ based on FODMAP content 2-week intervention, followed by reintroduction	Reduction in: Episodes/d of abdominal pain from 2 (IQR, 1.3–6.3) to 1.2 (0.4–3.3) (*p* = 0.024) Intensity of abdominal pain in VAS, 4.6 (2.5–6.4) to 1.4 (0.3–5.2); *p* = 0.035 Abdominal distension and gas No change: Stool characteristics	Less interference with daily activities Ease to follow diet: 6 = very easy, 7 = easy, 4 = little difficult, 3 = difficult	Small sample size, no control group
Brown 2020 New Zealand [33]	Retrospective observational	*n* = 29 4–17 years Chronic persistent and relapsing symptoms consistent with functional bowel disorder	Dietitian led education across all 3 phases of the diet	Complete resolution in: Bloating (92%), diarrhoea (87%), abdominal pain (87%) Symptoms from reintroduction: Fructans (67%), lactose (56%), polyols/fructose (7%), galacto-oligosaccharides (7%), sorbitol and fructose (24%)	62% written information easy to follow 79% diet easy to follow 31% not satisfied with improvement 59% not interested in changing diet further for symptom reduction	No control, retrospective symptom response survey taken 2–28 months after intervention
Dogan 2020 Turkey [34]	Randomised controlled trial	*n* = 60 (30 low FODMAP, 30 control) 6–18 years IBS (Rome IV)	Low FODMAP: <0.5 g FODMAP per meal vs. Control: ‘healthy’ diet avoiding certain foods specifically, e.g., caffeine, acidic, spicy, high fat 2-month intervention	Decrease in abdominal pain on 100 mm visual analogue scale after 2 months: Low FODMAP (3.8 ± 1.1 mm) vs. Control (2.0 ± 1.0 mm); *p* = 0.0001	Subjective ratings from doctors on symptom improvement using Clinical Global Impression Improvement—43% ‘much improved’ on low FODMAP vs. 3% control	No reporting of: Actual FODMAP intake Adherence or compliance to either diet.
Boradyn 2021 Poland [35]	Double-blinded randomised control trial	*n* = 27 5–12 years Functional abdominal pain (Rome III)	Modified NICE ^a^ guidelines (*n* = 14) vs. low FODMAP diet (*n* = 13) All food provided 1-month intervention	Abdominal pain intensity and frequency: No differences between diets (intensity, *p* = 0.167 and frequency, *p* = 0.332) Low FODMAP: tendency towards improvement NICE: reduction (*p* < 0.01) Stool consistency: No differences between groups	Higher non-compliance in 2nd week of low FODMAP diet (median: 6.0, IQR: 4.0–9.0 times per week) vs. NICE (median: 1.5, IQR: 1.0–2.0) (*p* < 0.05)	Both diets considered to be therapeutic—no placebo control

^a^ NICE, National Institute for Clinical Excellence.

**Table 2 nutrients-14-04369-t002:** Practical ways of keeping parents engaged when their child is prescribed a low-FODMAP diet.

Strategy	Examples
Ask	Gather information on the parents’ current understanding of the low-FODMAP diet, what they may have heard and assumptions that they have Ask about the entire family’s food preferences, i.e., picky eaters in siblings/food allergies to understand the dynamics within the family
Educate	Explain to parents that their child’s version of the diet may be different to what they have heard, and may potentially be more ‘gentle’ and tailored Remind parents of the short-term nature of strict restriction of FODMAPs, if this approach taken, and ultimate goal of finding the minimal level of dietary restriction needed for symptom control If higher discretionary foods are more prevalent in a child’s diet, provide parents with education regarding the benefits of removing/reducing these foods for overall long-term health benefits, eating habits and potentially improvement in symptoms
Tailor	Ensure that the dietary swaps suit the needs of not just the child with gastrointestinal symptoms, but are manageable for the parents with other competing siblings and priorities Ensure family-friendly recipes are provided that suit the individual needs of the family and do not require cooking multiple meals for different people Provide specific brands of lunch-box-friendly snacks specific to the region.

**Table 3 nutrients-14-04369-t003:** Behavioural and lifestyle considerations during consultations with paediatric gastrointestinal patients.

Age Group	Behavioural and Lifestyle Considerations
5–12 years	Parents have more control over food choices and food environment Establishing eating patterns Less privacy regarding toileting behaviours Participation in activities outside of the home, i.e., school camps Health literacy of family Meals provided by school or outside school hours care
12–17 years	Eating patterns and habits have become more established Increasing social pressures and spending more time away from home Pubertal growth and development Body image issues and risks of disordered eating habits. Gaining independence and may engage in more rebellious behaviours Autonomy with money and purchasing foods outside of the house More privacy when it comes to toileting behaviours Participation in activities outside of the home, e.g., school camps, spending nights away from home Health literacy of family Meals provided by school

**Table 4 nutrients-14-04369-t004:** Red flags to consider during assessment of the paediatric GI patient.

Red Flag Symptom	What Are We Missing?
Nocturnal waking symptoms	Gastroesophageal reflux disease (GORD)
Rectal bleeding	Inflammatory bowel disease (IBD), malignancy
Anaemia	Coeliac disease, IBD, malignancy
Suboptimal growth/unexplained weight loss	Coeliac disease, IBD, malignancy, non-IgE-mediated allergy
Recurrent vomiting	Infection, non-IgE mediated allergy
Family history of IBD/coeliac disease	Coeliac disease, IBD
Fever	Infection, malignancy, IBD
Coeliac disease not yet excluded	Coeliac disease
Pain in an unusual place (i.e., joint pain, difficulty swallowing, chest pain, etc.)	GORD, IBD, malignancy, eosinophilic esophagitis
Delayed puberty	Coeliac disease, IBD, malignancy

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
