# Peer review of "Application of The FODMAP Diet in a Paediatric Setting"

_nutrients, 2022, doi:10.3390/nu14204369_

Round 1
Reviewer 1 Report
Thank you to the authors for this nice well-written review paper.
The topic is important and there are little review papers to summarise using the low FODMAP diet in the paediatric setting.
Therefore, this is a necessary paper to be published.
I do not have major additions to the manuscript. I would love to have a table to show differences in approach between adults and paediatric patients. I would also love to have a segment on practical ways we can engage parents in order to make this intervention effective.
Author Response
We thank the reviewer for their positive comments and acknowledgement of the importance of this paper.
I would love to have a table to show differences in approach between adults and paediatric patients. I would also love to have a segment on practical ways we can engage parents in order to make this intervention effective.
In relation to the differences between paediatrics and adults, we feel that we have covered this enough in the paper. Additionally, there are certain adult population groups who may also benefit from using the FODMAP gentle proposed framework.
We have added in a table in engaging parents as per recommendations, please see below for table:
Table 2: Practical ways of keeping parents engaged when their child is prescribed a low FODMAP diet
Strategy |
Examples |
Ask |
· Gather information on the parents current understanding of the low FODMAP diet, what they may have heard/assumptions they have · Ask about the entire family’s food preferences, i.e., picky eaters in siblings/food allergies to understand the dynamics within the family |
Educate |
· Explain to parents that their child’s version of the diet may be different to what they have heard, may potentially be more ‘gentle’ and tailored · Remind parents of the short-term nature of strict restriction of FODMAPs, if this approach taken, and ultimate goal of finding the minimal level of dietary restriction needed for symptom control · If higher discretionary foods are more prevalent in a child’s diet, provide parents with education regarding the benefits of removing/reducing these foods for overall long-term health benefits, eating habits and potentially improvement in symptoms |
Tailor |
· Ensure the dietary swaps suit the needs of not just the child with gastrointestinal symptoms but are manageable for the parents with other competing siblings and priorities · Ensure family friendly recipes are provided that suit the individual needs of the family and do not require cooking multiple meals for different people · Provide specific brands of lunch box friendly snacks specific to the region. |
Reviewer 2 Report
Dear Authors
It was a pleasure to review this manusript.
yyou have compiled a good-quality review that includes a properly prepared introduction, individual points, the role of a dietitian and a summary.
All I can say is to extend the concluding part.
Tables and graphs should appear immediately after the mention, not at the end of the article.
I believe that the preparation of a systematic review by the authors would be more beneficial and would ensure a better quality of data
Author Response
We thank the reviewer for their positive feedback.
All I can say is to extend the concluding part
- We feel that if we extend the conclusion, there may be too much repetition in the paper.
I believe that the preparation of a systematic review by the authors would be more beneficial and would ensure a better quality of data
- In regard to a systematic review, while this would be valuable, there are limited publications at this point in time, thus we feel that a systematic review would be inappropriate.